# Research on Self-Stiffness Adjustment of Growth-Controllable Continuum Robot (GCCR) Based on Elastic Force Transmission

**DOI:** 10.3390/biomimetics8050433

**Published:** 2023-09-18

**Authors:** Mingyuan Wang, Jianjun Yuan, Sheng Bao, Liang Du, Shugen Ma

**Affiliations:** 1Shanghai Key Laboratory of Intelligent Manufacturing and Robotics, School of Mechatronic Engineering and Automation, Shanghai University, Shanghai 200444, China; wmy20008@163.com (M.W.); terabit@shu.edu.cn (J.Y.); 2Shanghai Robotics Institute, Shanghai University, Shanghai 200444, China; liang_du@outlook.com; 3Department of Robotics, Ritsumeikan University, 1-1-1 Nojihigashi, Kusatsu-Shi 525-8577, Japan; shugen@se.ritsumei.ac.jp

**Keywords:** growth-controllable continuum robot (GCCR), rod-driven continuum robot, stiffness adjustment mechanism (SAM), elastic force transmission, antagonism mechanism

## Abstract

Continuum robots have good adaptability in unstructured and complex environments. However, affected by their inherent nature of flexibility and slender structure, there are challenges in high-precision motion and load. Thus, stiffness adjustment for continuum robots has consistently attracted the attention of researchers. In this paper, a stiffness adjustment mechanism (SAM) is proposed and built in a growth-controllable continuum robot (GCCR) to improve the motion accuracy in variable scale motion. The self-stiffness adjustment is realized by antagonism through cable force transmission during the length change of the continuum robot. With a simple structure, the mechanism has a scarce impact on the weight and mass distribution of the robot and required no independent actuators for stiffness adjustment. Following this, a static model considering gravity and end load is established. The presented theoretical static model is applicable to predict the shape deformations of robots under different loads. The experimental validations showed that the maximum error ratio is within 5.65%. The stiffness of the robot can be enhanced by nearly 79.6%.

## 1. Introduction

### 1.1. Background and Previous Work

Industrial robots are provided with high rigidity to perform manufacturing tasks with accuracy, speed, and load capacity. These robots operate in confined spaces away from workers because of their use of inflexible linkages. Thus, continuum robots utilizing elastic components, such as series elastic actuators, have been developed to improve safety and allow for closer human–robot interaction in manufacturing environments.

Inspired by the biology of snakes, elephant trunks, and octopus tentacles, continuum robots are widely studied to reproduce similar functions [1,2]. Continuum robots (CRs) are manipulators that are continuously curved and flexible, making them a safe option for aero-engine blades [3,4], deep cavities [5,6], and human bodies [7]. Continuum robots, which can be wire-driven, rod-driven, or pneumatically controlled, have inherently robust structures that are lightweight and flexible. The slenderness ratio of continuum robots is very high [8]. Generally, the lower the slenderness ratio is, the larger the workspace and the lower the stiffness are [9]. Although their high flexibility is advantageous, it can lead to low accuracy and limited payload capacity. In 2018, a continuum robot named “LEeCh” inspired by real leeches was proposed for large deformation and extensive workspaces [10]. However, a tradeoff between workspace and load capacity should be considered first when designing high-performance continuum robots with a high slenderness ratio. Stiffness adjustment of the continuum robots is one feasible way to improve the performance of the continuum robots.

In recent years, researchers have developed several methods for adjusting the stiffness, as shown in Table 1. Mahvash et al. [11] proposed a control law that is based on an accurate approximation of a continuum robot’s coupled kinematic and static force model. Bajo et al. [12] presented a hybrid motion/force control method for sensing and control of multi-backbone continuum robots in a unified framework. Apart from methods based on algorithms, physical adjustment methods typically achieve a wider range of stiffness [12]. For most continuum robots actuated by air or fluid [13,14,15], granular jamming is a well-known variable stiffness mechanism widely applied, which involves an elastic membrane filled with small particles to enhance the robot body rigidity. This concept is uncomplicated and simple to construct but difficult in application. Functional materials also have been considered to realize stiffness adjustment features for the robot, such as magnetorheology and electrorheology materials [16], low-melting-point materials [17], and shape-memory alloys (SMA). Another practical methodology is antagonism. McKibben actuators [18] are a type of structure that can increase stiffness by coupling their extension and contraction. By varying the pressure applied to the actuator, the stiffness of the structure can be changed. However, the control can be complicated due to its complex structure.

In [19], recent advances, current limitations, and open challenges in the design, modeling, and control of continuum robots are discussed. In conclusion, there are four main ways to increase motion accuracy or achieve stiffness adjustment for continuum robots:Active stiffness adjustment: It requires an additional structure or mechanism to lock the robot actively, which can be understood as active stiffness adjustment;Passive stiffness adjustment: Based on the motion characteristics of the robot, stiffness adjustment is achieved without introducing additional actuation components;Lightweight design: The use of lightweight materials can effectively reduce the impact of robot weight on motion accuracy in gravity; continuum robots with gradually changing diameters can effectively shorten the distance between the center of mass and fixed end;Kinematics compensation: To fit the target by changing the actuation displacement initiative by kinematics algorithms.

For growth-controllable continuum robots, it is of great importance for them to have the capability of adjustable stiffness [20]. Due to the stretch-retractable motions of the GCCR directly affecting the change of its center of mass (CM), if the stiffness does not adjust accordingly, the consistency of the end motion accuracy cannot be guaranteed. Above all, passive stiffness adjustment has a unique appeal, as it requires no other drivers or mechanisms. Additionally, the GCCR happens to have the property of variable length and can serve as the only input.

Recently, researchers have focused their attention on nature-inspired designs [21,22] as well as on the studies of soft materials [23]. Bio-inspired solutions are widely adopted in different engineering disciplines. In nature, mollusks can change their body stiffness [24]. Earthworms are invertebrates, moving by means of muscular activity alone [25]. For example, earthworms change the tensile strength of their muscles by producing varying chamber pressures. When extending the body and digging holes in the soil, their body stiffness will further increase [26]. Although an earthworm mainly expands and contracts along the axial direction, its radial pressure is higher than the axial pressure, which means that it achieves stiffness changes by adjusting the radial pressure [27]. Indeed, for a GCCR using the pneumatic-driven method, adjusting the stiffness by changing the inflation pressure is a very convenient way [28].

Taking the continuum robot as a whole system, one method is to modify the internal forces between different components, such as the mechanisms of locking [29], and stiffness component insertion [9] SMA springs, and levers and cables were designed. In 2020, Yang et al. [29] established a static model based on virtual work and presented a novel variable stiffness mechanism powered by SMA. In 2022, Wang et al. [30] proposed a novel variable stiffness continuum robot using built-in winding ropes. By controlling the temperature of SMA, a large variable stiffness range of 300% of the robot is achieved. They increase stiffness by heating the SMA springs with a current, but these two types of robots require additional independent locking mechanisms.

This is a worthwhile study in a GCCR that uses variable length telescopic motion as the input and realizes self-stiffness adjusting. In this way, additional locking mechanisms are not necessary. Therefore, novel mechanisms are still required to achieve effective stiffness adjustment for a GCCR. Based on the comparison shown in Table 1, a locking mechanism is presented in this research to meet the requirements. Tension springs and a built-in cable-coupled mechanism were adopted in the robot and actuated by the length changing of the robot to achieve stiffness adjustment in a rod-driven GCCR. Because the locking mechanism is an elastic component built into the robot and related to the robot’s telescopic length, it has fewer restrictions and is beneficial for the rod-driven continuum robot with a growth control feature.

**Table 1 biomimetics-08-00433-t001:** Comparison of methods to enhance the stiffness of the continuum robot.

Reference	Stiffness Adjustment Method	Robot Length (m)	End Load (N)	Deflection (m)	Stiffness (N/mm)	Increase Percent (%)	Actuation for Stiffness Adjustment
Yang et al. [29]	Locking mechanism	0.88	3	0.135	0.037	187	Independent
Kang et al. [31]		-	-	-	-	-	Independent
Ours		0.2–0.75	5	0.018	0.032	80	Coupled
Kim et al. [15]	Layer jamming	0.44	3.8	0.02	0.198	90	Independent
Li et al. [32]		0.063	1.1	0.024	0.046	540	Independent
Wei et al. [33]	Particle jamming	0.15	3.5	0.014	0.259	900	Independent
Cianchetti et al. [14]		0.050	4.1	0.016	0.256	36	Independent
Kim et al. [34]	Antagonistic actuation	0.087	5	0.01	0.484	198	Independent
Zhao et al. [9]	Inserting rigid rod	0.315	-	-	2.71	983	Independent

### 1.2. Contribution

This paper describes modeling and experimental studies on a passive stiffness adjustment method for growth-controllable continuum robots. The main contributions are:A stiffness adjustment mechanism (SAM) is proposed and built in a growth-controllable continuum robot (GCCR) to improve the motion accuracy in variable scale motion.A statistics model that considers the weight of the robot and the end load is constructed and the shape of the robot can be predicted.Experimental testing is carried out to investigate the effect of the proposed SAM, modeling errors, and stiffness enhancement. The results provide efficacious insights to improve the design of the stiffness adjustment mechanism.

### 1.3. Outline

This paper addresses the passive stiffness adjustment feature for growth-controllable continuum robots and focuses on the following aspects: Section 2 briefly introduces the essential characteristics of a generalized growth-controllable continuum robot. In Section 3, the working principle and mechanical design of the stiffness adjustment mechanism (SAM) for the growth-controllable continuum robot are offered, and the relationship between robot telescopic motion and internal force transmission is analyzed. Section 4 establishes a static modeling of the continuum robot, which considers gravity and end load. In Section 5, the robot prototype and test platform are fabricated to verify the stiffness adjustment mechanism. Section 6 concludes the paper.

## 2. Generalized Growth-Controllable Continuum Robot

### 2.1. Configuration

A generalized design of a rod-driven continuum robot with growth-controllable feature is shown in Figure 1. This robot has a real or virtual bone and is composed of drive rods and *N* constraint disks. For the presented GCCR, there are three degrees of freedom, namely, bending, rotating, and retracting [35]. Based on the kinematics, the motion of the GCCR in space can be achieved by driving the drive rods collaboratively. Moreover, during bending or rotating motion, the GCCR can also engage in the stretching or retracting motion, as shown in Figure 1. In general, the basic kinematics of the continuum robot are based on the constant curvature arc assumption. Therefore, the kinematics of the GCCR are simplified as a constant curvature arc [1], while the arc supports the virtual bone of the GCCR.

### 2.2. Kinematics

The model and coordinate system of the GCCR are shown in Figure 2. As described in Ref. [1], the shape of the GCCR, which refers to virtual bone, is a constant curvature arc under the constraint of the serial drive rods and constraint disks. Thus, $\Phi =[\theta_{0},\phi ,L]$ could be defined as the posture parameters of the GCCR. According to the guided end position of the GCCR in the bending plane: (1)$$P_{bp}=\left[\frac{\left(1-cos\theta_{0}\right)}{\kappa },0,\frac{sin\theta_{0}}{\kappa }\right]$$
where $\kappa =\frac{\theta_{0}}{L}$ is the curvature of the continuum manipulator. In addition, for driving the GCCR, the relationship between the length of each fiberglass rod and the posture parameters are expressed as follows: (2)$$\Delta L_{i}=rcos\left(\frac{\pi (2i-1)}{3}+\phi \right)\theta_{0},i=1,2,3$$
where *r* is the distance between the drive rods and the plane that is perpendicular to the bending plane in the cross-section. For the presented GCCR, the extensible range of length is designed from 0.20 to 0.75 m.

### 2.3. Workspace

The workspace of a growth-controllable continuum robot reflects the motion capability, while the operating performance could be derived from the number of the degree of freedom. E. Amanov et al. [36] proposed a novel follow-the-lead tendon-driven continuum robot design, which features an additional degree of freedom in each robot section. A workspace volume increase of 22.5% compared with tendon-driven continuum robots with fixed section lengths is achieved [36]. Based on the forward kinematics, as in Equation (Equation 2), a traditional continuum robot with a constant-length backbone has two degrees of freedom. The position of the guided end can be obtained by the Monte Carlo method [37]. Thus, the inner space of the peach-shaped surface is inaccessible for the traditional continuum robots, as shown in Figure 3a,c. In comparison to traditional continuum robots, the reachable workspace of the GCCR is shown in Figure 3b,d, which is a three-dimensional cavity with thickness instead of a surface.

## 3. Stiffness Adjustment Mechanism (SAM)

A typical characteristic that sets the GCCR apart from traditional continuum robots is length changing. As stiffness adjustment of the GCCR is desired, the length change of the robot can be considered as the only independent variable. In this section, a novel mechanism based on spring and cable is proposed and built into the GCCR for its passive adjustment of stiffness. Based on the generalized rod-driven GCCR structure, a stiffness adjustment mechanism was proposed that can change the internal frictional force between the drive rods and constraint disks as the robot length changes. This design uses anisotropic distributed cables and geometry relationships to achieve local frictional force, unlike the antagonistic actuators [38], central-cable-tensioning methods [39], or other mechanisms actuated by additional actuators [29,30].

### 3.1. Working Principle

The configuration of the SAM built in a GCCR is shown in Figure 4. This robot is composed of three drive rods and three groups of constraint disks, which are evenly distributed in circumferential directions. Compared with the tendon-driven continuum robot proposed in [36], the rod-driven continuum robots require no real backbone to sustain the body, thus securing internal space for other parts or sensors inside the robot body. Three cables connected with three tension springs run through the constraint disks and are fixed on the guided end to provide tension force to each cable. The drive rods and cables are actuated to bend, which are connected without twist torque to the guided end and move freely among the constraint disks. The compression springs (shown in yellow) installed on the drive rods are utilized to maintain the relative distance of the constraint disks when the robot is actuated. Therefore, when the robot body carries out stretch-retractable motions, it directly affects the tension force of the spring. Thus, tension forces $F_{c}$ are applied to the two ends of the cable.

The three cables inside the robot traverse constraint disks at different positions, as shown in Figure 4b,c. When a tension force is generated by the tension spring, the direction of the resultant force generated by the three cables on each constraint disk is different; $S_{i}$ represents the direction of the resultant force applied to each constraint disk. Therefore, the frictional force between the drive rods and constraint disks increases. This mechanism enhances the internal frictional force inside the robot and limits the rod displacement relative to the constraint disks.

Tension springs are utilized to pull the cables, and compression springs are utilized to maintain the relative position of the constraint disks due to their high power–weight ratio, flexibility, stable force deformation relationship, and compactness making them suitable for implementation in such a slender continuum robot. Thus, the stiffness adjustment of the robot can be achieved by the length changing of the robot. In this system, the only variable that affects the stiffness of the robot is the robot’s length.

As a result, when the total length of the robot is short, the springs and cables provide a small tension force applied to the constraint disks and produce small frictional forces; when the robot is at a long length, the springs and cables are tightened, and the friction force increases. In this process, it is tensioned, and it will apply a normal force $F_{n}$ to every constraint disk. So, there is a trend that the robot will bend under the effect of gravity. The friction between constraint disks and drive rods will provide resistance force, which means the overall stiffness of the continuum robot changes.

### 3.2. Force Transmission

Based on the working principle of the stiffness adjustment mechanism, a force transmission chain can be obtained, as shown in Figure 5. When the stiffness of the proposed continuum robot changes, new variables, such as the tension of the springs and the resulting frictional force, will be introduced. The normal force $F_{n}$ acting on each constraint disk can be expressed as
(3)$$F_{n}=\frac{d_{2}-d_{1}}{3\sqrt{{\left(\frac{\Delta l_{s}}{4}\right)}^{2}+{\left(d_{2}-d_{1}\right)}^{2}}}F_{c}$$
where $\Delta l_{s}$ is the length changing of the cable. $d_{1}$ and $d_{2}$ are the distances from the centerline of the contact position between the cable and the constraint disk, respectively. $F_{c}$ is the tension force of the tension spring, which can be expressed as
(4)$$F_{c}=k_{s}\Delta l_{s}$$
where $k_{s}$ is stiffness of the tension spring and $\Delta l_{s}=\left(L_{2}^{2}-L_{1}^{2}\right)/3$, $L_{1}$ and $L_{2}$ are the lengths of the robots before and after the change, respectively.

Thus, the frictional friction provided by the cable is
(5)$$F_{f}=\mu F_{c}$$
where μ is the frictional coefficient of cable and constraint disks. Due to the slender structure of the robot, assuming its mass distribution is uniform, the gravity it receives can be equivalent to a cantilever beam. As a result, the stiffness adjustment mechanism provides a compensated torque of $M_{s}=3F_{f}r$ to resist gravity.

## 4. Static Modeling and Analysis

For solving the static deformation of GCCR, the first step is to create an equivalent fixed-end beam model. The virtual bone of the continuum robot is parallel to each drive rod due to the constraint disks, which keeps the spacing of each drive rod consistent. Thus, three assumptions are made as follows:The continuum robot has a slender structure, and the mass distribution is assumed uniform;Each curve of the drive rod is parallel to each other, including the virtual bone;Due to the compression springs being utilized to keep an equidistant sate of the constraint disks, the elastic potential energy is negligible.

### 4.1. Elastic Potential Energy and Bending Stiffness

As for the structure of the GCCR, the fiberglass rods are parallel to each other when the continuum robot is deformed. The elastic potential energy of each fiberglass rod can be expressed as
(6)$$U_{i}=\int_{0}^{L_{i}}\frac{E_{i}I_{i}\kappa_{i}{\left(s_{i}\right)}^{2}}{2}ds$$
where $\kappa_{i}\left(s_{i}\right)$ represents the curvature of each point on the fiberglass rod; $E_{i}I_{i}$ is the bending stiffness of the fiberglass rod; $L_{i}$ is the length of each fiberglass rod.

According to the isometric line principle, the equivalent bending stiffness is obtained as
(7)$$E_{e}I_{e}=3E_{i}I_{i}$$

Hence, the equivalent-bending stiffness $E_{e}I_{e}$ of the continuum robot is determined based on Equation (Equation 7). Subsequently, a large deflection static model can be conducted with the structure parameters.

### 4.2. Static Modeling

Figure 6 shows the static analysis of the continuum robot. According the Bernoulli–Euler beam theory [40], the bending moment at point *B* can be written as
(8)$$M_{b}=P_{s}\left(b-y\right)+M_{s}$$
where $M_{b}$ is the bending moment acting on the guided end of the GCCR. The equivalent mass $P_{s}=mg/2$ of the robot can be defined by the product of the weight per unit length and the elongation of the robot as
(9)$$m=\sum_{i=1}^{3}\rho_{i}l_{i}=\Xi L_{i}$$
where Ξ represents the equivalent density of the robot, including the weight of constraint disks and tension springs.

Further, the curvature *K* at point *B* is
(10)$$K=\frac{d\theta }{ds}=\frac{M_{b}}{E_{e}I_{e}}=\frac{P_{s}\left(b-y\right)+M_{s}}{E_{e}I_{e}}$$

Subsequently, differentiating Equation (Equation 10) respect to *s* yields
(11)$$\frac{d^{2}\theta }{ds^{2}}=\frac{P_{s}}{E_{e}I_{e}}\left(-\frac{dy}{ds}\right)$$
where ds, dx, and dy can be considered infinitesimal; thus, the relationship between the three infinitesimals is by trigonometric functions, which can be expressed as $d_{x}/d_{s}=sin\theta$, $d_{y}/d_{s}=cos\theta$.

Subsequently, Equation (Equation 11) is transformed as
(12)$$\frac{d^{2}\theta }{ds^{2}}=\frac{P_{s}}{E_{e}I_{e}}cos\theta$$

The rewritten left side of Equation (Equation 12) is as follows: (13)$$\frac{d^{2}\theta }{ds^{2}}=\frac{d}{d\theta }\left(\frac{K^{2}}{2}\right)$$

Substituting Equation (Equation 12) into Equation (Equation 13) and integrating, the following equation can be obtained: (14)$$\int -\frac{P_{s}}{E_{e}I_{e}}cos\theta =\int \frac{d}{d\theta }\left(\frac{K^{2}}{2}\right)\to -\frac{P_{s}}{E_{e}I_{e}}sin\theta +c_{1}=\frac{K^{2}}{2}$$
where $c_{1}$ is the integral constant that can be obtained based on the boundary
(15)$${K|}_{s=L}=\frac{d\theta }{ds}{|}_{\theta =\theta_{0}}\to c_{1}=\frac{K^{2}}{2}+\frac{P_{s}}{E_{e}I_{e}}sin\theta_{0}$$

Thus, Equation (Equation 10) can be expressed as
(16)$$K^{2}=\frac{2P_{s}}{E_{e}I_{e}}\left(-sin\theta +sin\theta_{0}+\frac{M_{s}^{2}}{2P_{s}E_{e}I_{e}}\right)$$

Therefore,
(17)$$K=\frac{d\theta }{d_{s}}=\sqrt{\frac{2P_{s}}{E_{e}I_{e}}}\sqrt{\left(-sin\theta +sin\theta_{0}+\frac{M_{s}^{2}}{2P_{s}E_{e}I_{e}}\right)}$$

Integrating Equation (Equation 17), the length of the robot can be expressed as
(18)$$L=\int_{0}^{L}ds=\sqrt{\frac{2P_{s}}{E_{e}I_{e}}}+\int_{0}^{\theta_{0}}\frac{cos\theta d\theta }{\sqrt{\left(-sin\theta +sin\theta_{0}+\sigma \right)}}$$

Substituting $\alpha =\sqrt{\frac{P_{s}L^{2}}{E_{e}I_{e}}}$ into Equation (Equation 18), α is written as
(19)$$\alpha =\int_{0}^{\theta_{0}}\frac{d\theta }{\sqrt{\left(-sin\theta +sin\theta_{0}+\frac{M_{s}^{2}}{2P_{s}E_{e}I_{e}}\right)}}$$

Based on the above derivation, the guided end coordinates can be expressed as
(20)$$\left\{\begin{array}{c} \frac{a}{L}=\frac{1}{\alpha \sqrt{2}}+\int_{0}^{\theta_{0}}\frac{sin\theta d\theta }{\sqrt{\left(-sin\theta +sin\theta_{0}+\sigma \right)}}\\ \frac{b}{L}=\frac{1}{\alpha \sqrt{2}}+\int_{0}^{\theta_{0}}\frac{cos\theta d\theta }{\sqrt{\left(-sin\theta +sin\theta_{0}+\sigma \right)}} \end{array}\right.$$
where $\frac{a}{L}$ and $\frac{b}{L}$ are the dimensionless coordinates of the guided end along the *x*-axis and *z*-axis, respectively. $\sigma =\frac{M_{s}^{2}}{2P_{s}E_{e}I_{e}}$ is the load ratio.

### 4.3. Predicted Robot Shape

Based on the consumption, the shape of the continuum robot is a constant curvature arc. As shown in Figure 7, the bending angle of the fixed end (point B) is always unchanged regardless of the shape deformed owing to the different external load. When an external load and moment are acting on the deformed continuum robot, the known parameters are $P_{s}$, and $\theta_{0}$. Thus, the guided end coordinates $\left(L_{a}/L,L_{b}/L\right)$ can be calculated. By Equation (Equation 20), the starting point (point B) of the arc length and the guided end (point A) coordinates are known, and the center of the arc can be determined and depicted in the coordinate system as
(21)$$x^{2}+{\left(y+\sqrt{{\left(L_{a}/L\right)}^{2}+{\left(L_{b}/L\right)}^{2}}\right)}^{2}={\left(L_{a}/L\right)}^{2}+{\left(L_{b}/L\right)}^{2}$$

By intercepting the corresponding arc through $\theta_{0}$ or *L*, the approximate shape of the robot can be obtained.

## 5. Experiments and Results

### 5.1. Robot Prototype and Test Platform Setup

Based on the working principle and mechanical design mentioned above, a prototype is designed, as shown in Figure 8. The prototype consists of a growth-controllable continuum robot with a stiffness adjustment mechanism and a linear actuation module of three lead screws. The robot has a maximum length of 0.75 mm and an outer diameter of 0.1 mm, which is hollow without any cover for weight loss. Three fiber glass rods are assigned as the drive rods, which are fixed on the linear actuators. To reduce the robot’s weight, the constraint disks are made using 3D printing technology. The built-in cables of the robot are manufactured from fine-ground steel. The geometric and material parameters of the presented continuum robot are detailed in Table 2.

The test platform aims to tune the stiffness of the main body of the robot in a gravity field when the robot has a different length or with a different end load. A stiffness test bench is established to fit the relationship between the tension force and deformation of the tension springs. Since the virtual bone cannot be directly measured, one side of the midpoints of the robot is marked in contrary colors, and the midpoint can be recognized as the center line of the virtual bone. The shape of the robot was captured by digital camera and measured in Kinovea software [41].

### 5.2. Predicted Robot Shape in Different Actuation Displacement

In this subsection, the performance of the GCCR with SAM under different robot lengths is shown. In this case, the robot length ranges from 0.2 m to 0.75 m. To compare the results under different actuation conditions more clearly, the robot only bends in the *x*-*z* plane with no end load. In this paper, two types of errors are considered. One is the absolute errors (unit: m), representing the distance between two guided end coordinates in experiments and simulation. For example, $EE_{d}$ is the absolute error of the guided end, and $AE_{d}$ is the average absolute error of the guided end. The other is relative errors (unit: %), which is defined as the ratio of an absolute error to the robot length. For example, $EE_{L}=EE_{d}/L$ is the guided end error relative to robot length, and $AE_{L}=AE_{d}/L$ is the average guided end error relative to robot length.

The results are shown in Figure 9. In a gravity field, the deflection of a robot increases with the length of the robot, which means that the longer the continuum robot length, the smaller its stiffness. In addition, the predicted robot shape results obtained by the predicted model are matched with the experimental results. Table 3 presents the error analysis of the predicted and experimental results under different actuation displacements. The absolute errors and relative errors of the static model both increase with the elongation of the robot and show a gradually expanding trend. The maximum absolute error occurs when the robot has the longest length, which is 0.042 m. The relative error with the length of the robot is 5.57%.

Figure 10 shows the relationship between errors and robot length. From the results, the following results can be obtained:Both absolute and relative errors increase with the elongation of the robot.In general, the average error is greater than the robot end error.The average error almost shows a linear trend after the robot length is greater than 0.3 m, while the trend of end error change is not as significant as the average error.

### 5.3. Effect of Stiffness Adjustment Mechanism

In this case, the role of the stiffness adjustment mechanism is confirmed. Figure 11 shows the predicted robot shape and experimental results with SAM or not. It can be noted from the above two cases that the robot shape of the simulations is close to the experimental tests, but there still are some position errors, especially for the guided end. Table 4 presents the error analysis. Firstly, it can be seen that the SAM has little impact on the accuracy of the static model, indicating that the established force transmission models of cables and tension springs did not introduce significant errors into the original GCCR static model.

Secondly, $CD_{p}$ and $CD_{e}$ are the compensation distances for the guided end in the predicted model and in the experiments. The results show that simulation and experiments have improved end accuracy by 12.91% and 9.27%, respectively, indicating the effectiveness of SAM in improving GCCR motion accuracy.

### 5.4. Effect of Variable Stiffness Demonstration and Validation

As mentioned in Section 3, the stiffness of the continuum robot can be adjusted by changing the internal friction, whose relationship can be calculated by the static model with different actuation displacement inputs. Furthermore, the quotient between the end load $F_{cp}$ (N) and the corresponding deflection ΔX (m), as shown in the Equation (Equation 22), is used to characterize the stiffness $k_{r}$ of the continuum robot:(22)$$k_{r}=\frac{F_{cp}}{\Delta X}.$$

According to Equations (Equation 4) and (Equation 5), the maximum frictional force depends on the length changing of the tension springs. Thus, the deformations of the continuum robot obtained by experiments and simulations under the same end load 0.5 N, but different lengths are shown in Figure 12. Sta. (0.5 N, 0.2 m) is the initial state. Then, when the robot extends and the length becomes 0.35 m, the state changes to Sta. (0.5 N, 0.35 m). The deflections of the robot are collected from 0.2 m to 0.75 m in length and compare the slope of linear fitting between the presence and absence of SAM assistance. It can be seen that the rate of change in both conditions is close to linear. As shown in Figure 12b, the robot under the action of SAM has larger stiffness at the same robot length.

Keeping the condition as Sta. (0 N, 0.75 m), the magnitude of the end load is changed from 0 N to 0.5 N and the relationship of end load $F_{cp}$ and deflection in *x*-axis (ΔX) is shown in Figure 13. Under this condition, an approximate linear stiffness $k_{r}$ can be observed when the end load is bigger than 0.1 N. When the robot length is 0.75 mm, the stiffness of the robot with SAM and without SAM is 3.18 N/m and 1.77 N/m, respectively. Compared with a generalized continuum robot, the proposed stiffness adjustment mechanism can increase the stiffness of the robot by 79.6%.

Hysteresis can have an impact on the repetitive positioning accuracy of robots during actual tasks [42]. Figure 13 shows a circle of loading and unloading. In terms of absolute deflection, a hysteresis behavior is observed in the GCCR with SAM: there is a large hysteresis (up to 26.3 mm) between loading and unloading when the robot length is up to 0.75 m at a maximum load of 0.5 N. However, without SAM, the hysteresis behavior of GCCR is not so significant (5.2 mm) because locking mechanisms can cause an increase in internal friction, creating resistance to the robot’s recovery to the initial state.

### 5.5. Effect of Basic Motions on Cable Length Changing

Based on the theoretical derivation in Section 4, this subsection explains the influence of different robot motions on the tension force of the build in cable through mathematical derivation. According to Section 3, a generalized growth-controllable continuum robot has three basic motions: rotating, bending, and stretching. Therefore, there are three situations to discuss:

(1) When the robot performs stretching motion, as shown in Figure 14a,b, the relationship between the built in cable length $l_{dc}$ and the robot length (virtual bone length) changing ΔL can be expressed as
(23)$$l_{dc}=2\sqrt{{\left(\frac{L+\Delta L}{4}\right)}^{2}+{\left(2d\right)}^{2}}+2\sqrt{{\left(\frac{L+\Delta L}{4}\right)}^{2}+d^{2}}$$
where the length changing range is from *L* to 2L, and the initial length L=0.3 m.

Rotating as a robot bends at different ϕ is shown in Figure 1. Therefore, these two types of bending situations are discussed, one is ϕ=0, and the other is ϕ=π/2.

(2) When ϕ=0, during the bending motion of the robot, the length changing of the built in cable can be represented by
(24)$$l_{dc}=6\sqrt{{\left(\frac{b-a}{2}\right)}^{2}+4r^{2}}$$
where $a=2\left(\frac{L}{2sin(\frac{\theta_{0}}{4})}-r\right)sin\theta_{0}$ is the short side of the trapezoid, $b=2\left(\frac{L}{2sin(\frac{\theta_{0}}{4})}+r\right)sin\theta_{0}$ is the long side of the trapezoid, and *r* is the robot radius. $\theta_{0}$ ranges from 0 to π/4.

(3) When ϕ=π/2, as shown in Figure 14c, during the bending motion of the robot, the length changing of the built in cable can be represented by:(25)$$l_{dc}=\frac{4L}{sin\theta_{0}}sin\left(\frac{\theta_{0}}{4}\right)$$

The relationships between robot length changing and built in cable length changing are shown in Figure 14e–g. As shown in Figure 14e, when the robot only performs stretching motion, the length of the built-in cable shows an almost linear relationship in the current robot geometry parameters, and the average rate of cable length change is 87.1%. This result also indicates that using linear springs can achieve the desired stiffness change results.

Figure 14f,g show that the change in cable length is non-linear to the angle of binding, and the larger the angle, the greater the rate of change. When the robot performs bending motion in a vertical plane, the average rate of change within the range of 0 to π/4 is 18.9%, while the average rate of change within the range of 0 to π/8 is 7.7%; when the robot performs bending motion in a horizontal plane, the average rate of change within the range of 0 to π/4 is 10.4%, while the average rate of change within the range of 0 to π/8 is only 2.1%. Above all, compared to the rate of change in cable length during robot stretching motion, pure bending motions of the robot will not significantly affect the length change of the built-in cables. In other words, the significant change in the stiffness of the robot is mainly affected by the change in robot length.

## 6. Discussion and Conclusions

For most slender continuum robots, e.g., the follow-the-lead continuum robot proposed in [36], mass distribution and body weight do influence the motion accuracy and limit the load capacity. Thus, this paper introduced a stiffness adjustment mechanism (SAM) based on built in tension springs and cables, for passively adjusting the stiffness for a generalized growth-controllable continuum robot (GCCR). With a simple structure, the combination of cable and tension spring introduces a scarcity effect on the weight and mass distribution of the robot. Compared to the stiffness adjustment methods of using the active locking mechanisms mentioned in [29,31,36] and Table 1, the most attractive point is that the SAM requires no additional actuators, and instead utilizes the length change of the robot as the only input.

In this system, the length changing of the continuum robot directly influences the tension forces of the springs and produces internal frictional force to “lock” the drive rods. In modeling, a static model was established, which considered the continuum robot as an equivalent-end beam with the corresponding equivalent bending stiffness. The experimental validations showed that the maximum error ratio is within 6.82%. The stiffness of tension springs is a crucial factor in the ability of stiffness adjustment. For now, considering the expansion ratio and linear extension range, a tension spring stiffness of 43.8 N/m is selected, and a stiffness enhancement ratio of 79.6% is obtained. As analyzed in Section 5.5, different basic motions affect the length of the built-in cables, and it further affects the stiffness. It also explains that if the robot length or bending angle is large, the absolute error between the predicted model and experimental results will increase (Figure 9 in Section 5.2).

Additionally, combined with Figure 9 and Figure 11, it can be observed that the guided end errors are lower than the average errors of the overall shape of the robot. The maximum error of the robot is 6.82%, which is the average error of the robot shape without SAM, while the guided end error under the same conditions is only 3.03%. The reason is that the shape prediction method is based on the piecewise constant curvature kinematics (PCCK) method, simplifying the parallel structure of a continuous robot. This method is not inherently feasible and leads to larger shape deviations [36,43].

For future research, using a more accurate model (e.g., ROM, Cosserat) [44] will improve the average accuracy of the robot. This method of stiffness adjustment involves approximately linearly changing the normal force of the cable $F_{n}$ on the constraint disk, which is a type of locking mechanism. However, it is subject to resistance from static internal frictional force during the dynamic motion, resulting in hysteresis behavior and frustration. To solve this problem, future research may tend to decrease the internal friction. Compartmentalization is a design philosophy and has been successfully used in real constructions, the next-generation smart structures should be able to change their stiffness and connectivity [45]. Further, the proposed stiffness adjusting method here could be extended to other types of continuum robots, regardless of the actuation methods.

## 7. Patents

Mingyuan Wang et al., A driving component for a continuum robot, Chinese patent, ZL202111404400.5, Shanghai University, 2022. 12. 27. (Granted)Mingyuan Wang et al., A Modular Continuum Robot with Multiple Operation Modes, Chinese patent, ZL202111403853.6, Shanghai University, 2023. 8. 11. (Granted)

## Figures and Tables

**Figure 1 biomimetics-08-00433-f001:**
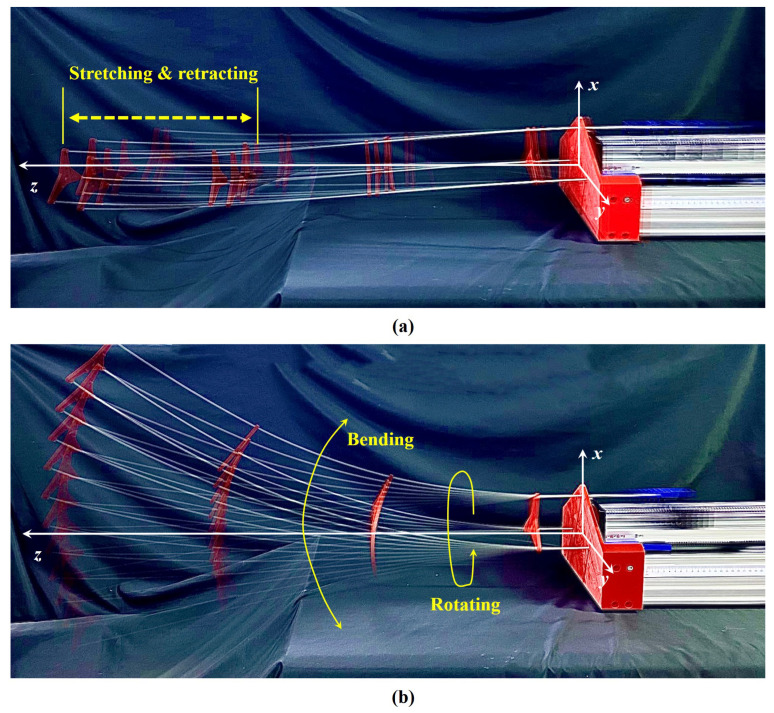
The realizable motions of a growth-controllable continuum robot: (**a**) stretch-retractable motion and (**b**) bending motion and rotating motion.

**Figure 2 biomimetics-08-00433-f002:**
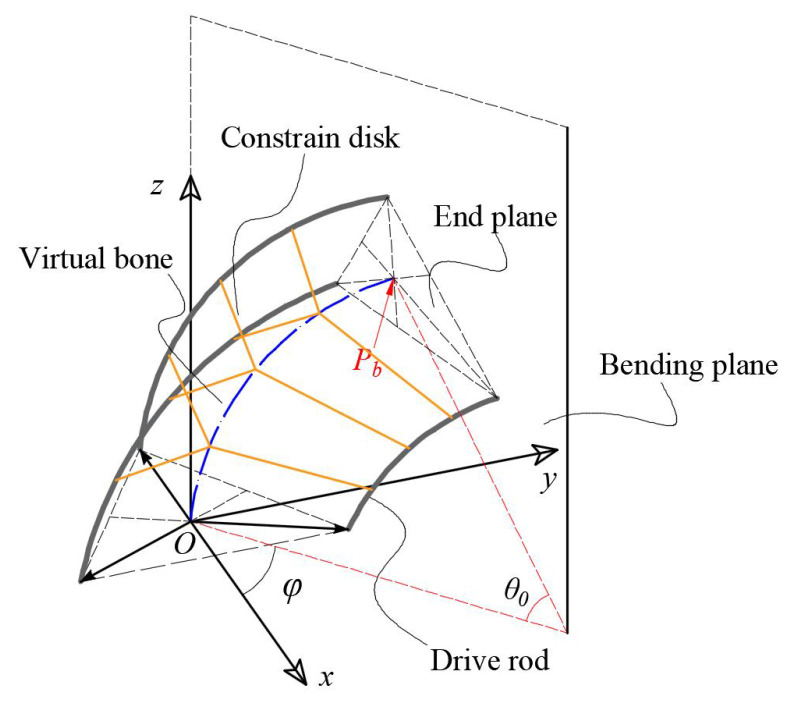
Kinematics nomenclature of a growth-controllable continuum robot.

**Figure 3 biomimetics-08-00433-f003:**
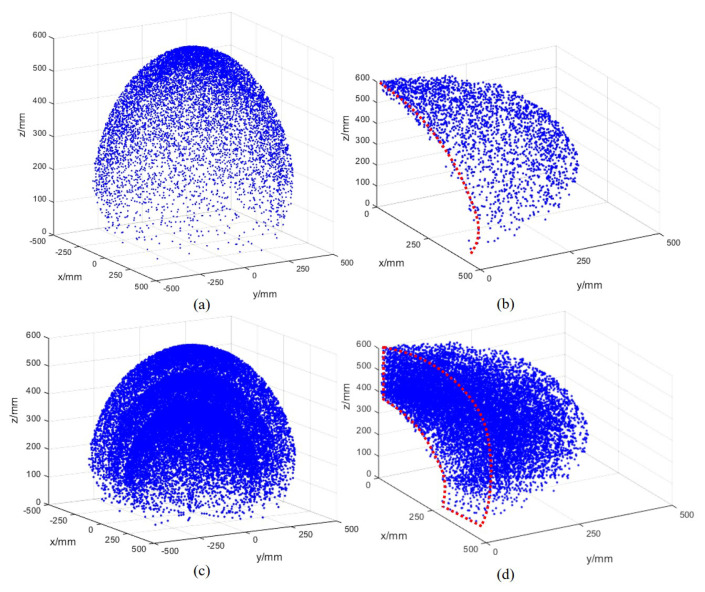
Simulation of workspace of the traditional continuum robot and the growth-controllable continuum robot (GCCR): (**a**) the workspace of a traditional continuum robot with the range of 0 < ϕ < 2π, 0 < $\theta_{0}$ < π/2, *L* = 600 mm, (**b**) quarter-section view of (**a**), (**c**) the workspace of a growth-controllable continuum robot with the range of 0 < ϕ < 2π, 0 < $\theta_{0}$ < π/2, 400 mm < *L* < 600 mm, and (**d**) quarter-section view of (**c**).

**Figure 4 biomimetics-08-00433-f004:**
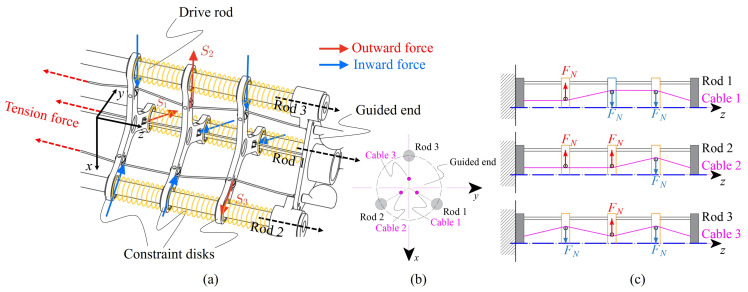
Conceptual schematic structure of the proposed continuum robot: (**a**) the growth-controllable continuum robot (GCCR) with proposed stiffness adjustable mechanism (SAM). $S_{i}\left(i=1,2,3\right)$ represents the direction of the combined force of the three cables on each constraint disk. (**b**) The arrangement of rods and cables of the robot for the guided end and (**c**) the details of routing of rods and cables through each constraint disk in each group.

**Figure 5 biomimetics-08-00433-f005:**
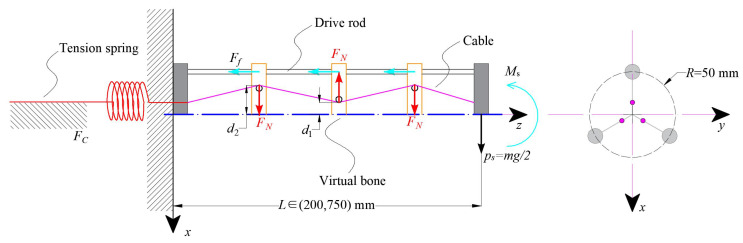
Force transmission of the stiffness adjustment mechanism (SAM) based on cables and tension springs.

**Figure 6 biomimetics-08-00433-f006:**
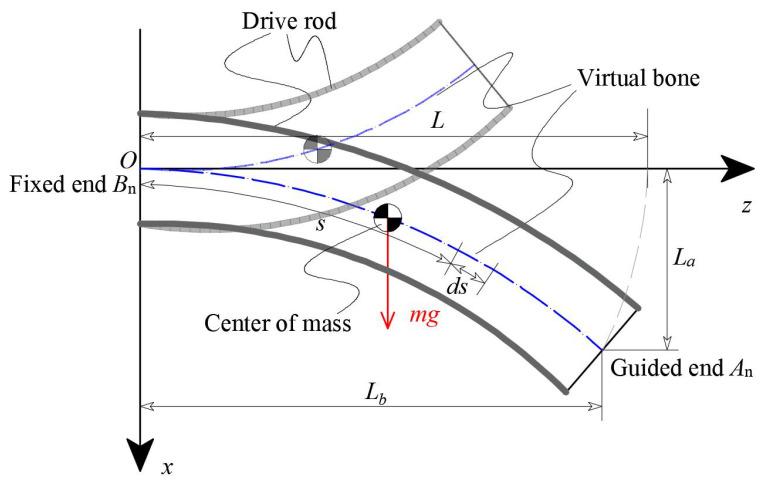
Static analysis of the continuum robot. The robot length *L* ranges from 200 mm to 750 mm.

**Figure 7 biomimetics-08-00433-f007:**
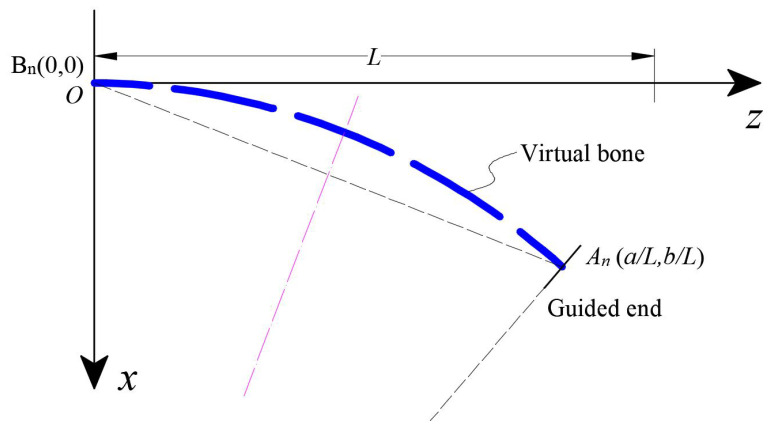
Predicted shape of the continuum robot.

**Figure 8 biomimetics-08-00433-f008:**
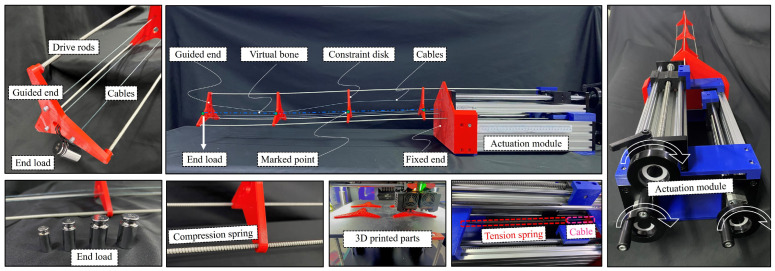
Prototype of the proposed growth-controllable continuum robot and test platform.

**Figure 9 biomimetics-08-00433-f009:**
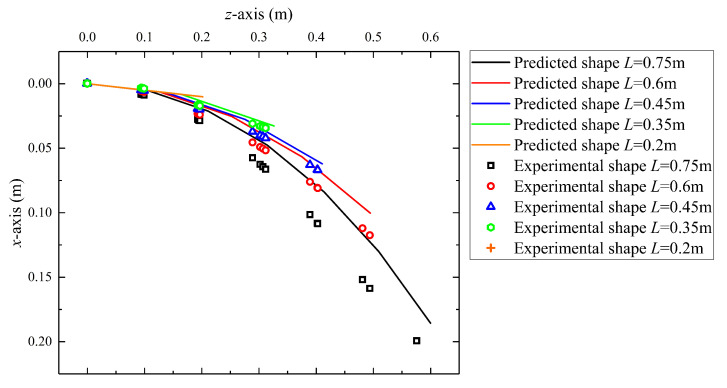
Predicted and experimental shape of the robot under different actuation displacement.

**Figure 10 biomimetics-08-00433-f010:**
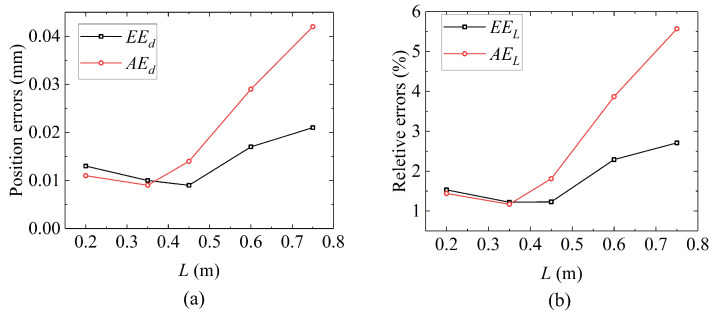
The relationship between errors and robot length: (**a**) the relationship between absolute errors and robot length and (**b**) the relationship between relative errors and robot length.

**Figure 11 biomimetics-08-00433-f011:**
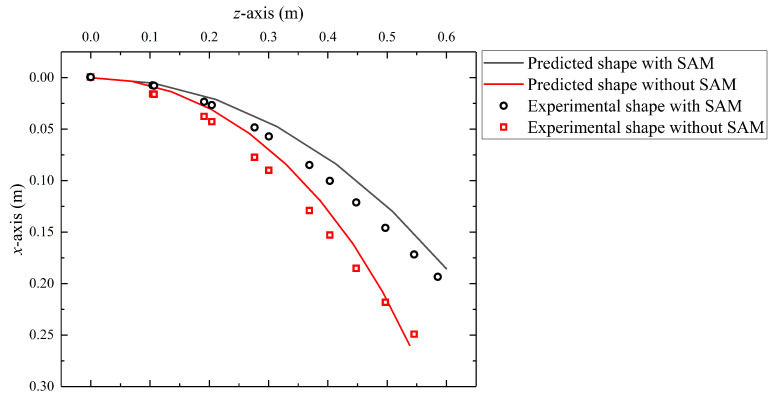
The effect of the stiffness adjustment mechanism.

**Figure 12 biomimetics-08-00433-f012:**
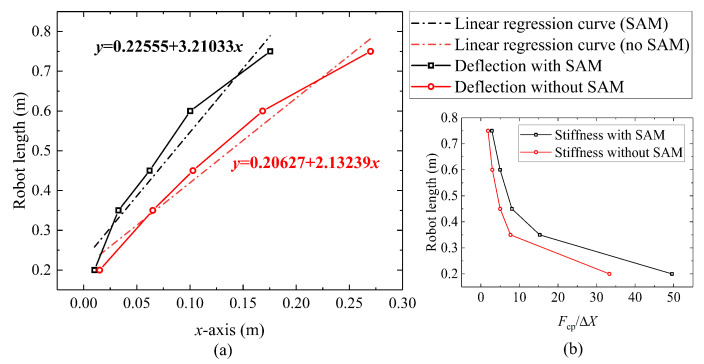
Effect of robot length on deflection: (**a**) relationship between the robot length and guided end deflection in *x*-axis and (**b**) stiffness comparison of the robot in different lengths.

**Figure 13 biomimetics-08-00433-f013:**
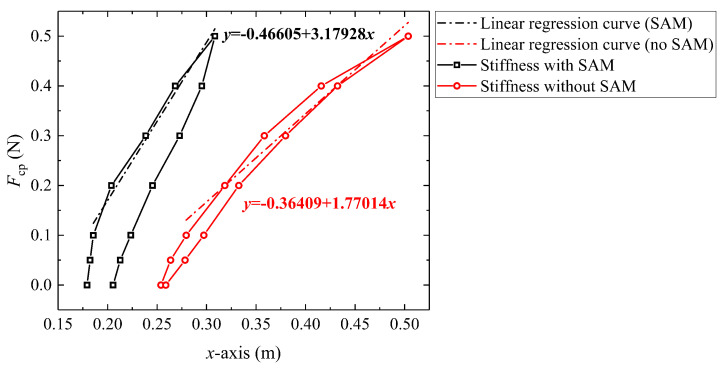
Relationship between the end load and guided end deflection in *x*-axis.

**Figure 14 biomimetics-08-00433-f014:**
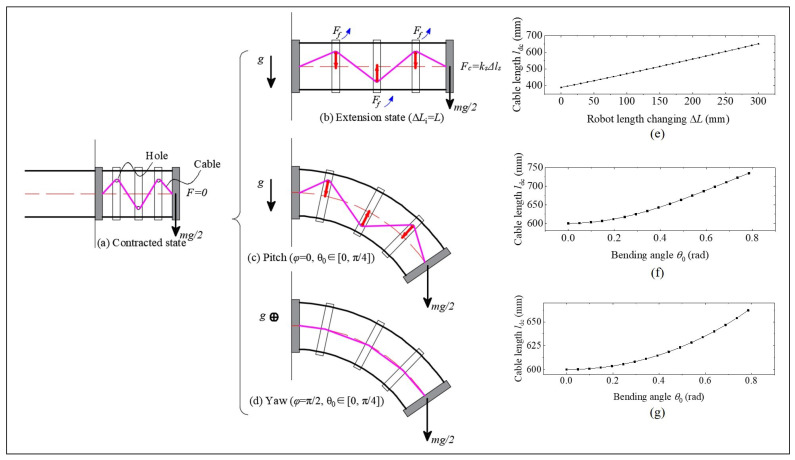
Effect of different motions on tension force: (**a**) a contracted state of the continuum robot. (**b**) Extension state of the continuum robot, at which the maximum length of the robot reaches twice its original length. Also, there are two pure bending motions, in which (**c**) is the robot performing bending motion in a horizontal plane, and (**d**) is the robot performing bending motion in a vertical plane. (**e**–**g**) are the relationship of built-in cable length and robot length changing or bending angle.

**Table 2 biomimetics-08-00433-t002:** Material and geometric parameters of the prototype.

Variable	Value	Unit	Description
*n*	3	—	The number of drive rods
$L_{0}$	0.2	m	The initial length of the GCCR (minimum length of the robot)
*L*	0.2∼0.75	m	The length range of the GCCR in this paper
$E_{i}$	7.5 × 10${}^{-10}$	Pa	Young’s modulus of fiberglass
$I_{i}$	2.029 × 10${}^{-8}$	kgm${}^{2}$	The moment of inertia of an area of fiberglass
Ξ	0.17	kg/m	The equivalent density of the GCCR
*r*	0.05	m	The distance between the fiberglass rods and the virtual bone
$r_{g}$	0.0015	m	Diameter of each fiberglass rod
*g*	9.81	m/s${}^{2}$	Gravitational acceleration
$k_{s}$	43.8	N/m	The stiffness of the tension spring
μ	0.128	—	Frictional coefficient of cable and constraint disks

**Table 3 biomimetics-08-00433-t003:** Error analysis of predicted and experimental results under different actuation displacement.

Length (m)	State	${\mathrm{EE}}_{d}$ (m)	${\mathrm{EE}}_{L}$ (%)	${\mathrm{AE}}_{d}$ (m)	${\mathrm{AE}}_{L}$ (%)
0.20	With SAM *	0.013	1.44	0.011	1.53
0.35		0.010	1.17	0.009	1.22
0.45		0.009	1.23	0.014	1.81
0.60		0.017	2.29	0.029	3.87
0.75		0.021	2.71	0.042	5.57

* SAM: stiffness adjustment mechanism. $EE_{d}$ is the absolute error of the guided end, $EE_{L}=EE_{d}/L$ is the guided end error relative to robot length, $AE_{d}$ is the average absolute error of the guided end, $AE_{L}=AE_{d}/L$ is the average guided end error relative to robot length.

**Table 4 biomimetics-08-00433-t004:** Error analysis of predicted and experimental results under with SAM or not.

Length (m)	State	${\mathrm{EE}}_{d}$ (m)	${\mathrm{EE}}_{L}$ (%)	${\mathrm{AE}}_{d}$ (m)	${\mathrm{AE}}_{L}$ (%)	${\mathrm{CD}}_{p}$ (m) [${\mathrm{CD}}_{\mathrm{pL}}$ (%)]	${\mathrm{CD}}_{e}$ (m) [${\mathrm{CD}}_{\mathrm{eL}}$ (%)]
0.75	With SAM *	0.021	2.71	0.042	5.56	0.097 [12.91]	0.069 [9.27]
	No SAM	0.023	3.03	0.051	6.82		

* SAM: stiffness adjustment mechanism. $EE_{d}$ is the absolute error of the guided end, $EE_{L}$ is the guided end error relative to robot length, $AE_{d}$ is the average absolute error of the guided end, $AE_{L}$ is the average guided end error relative to robot length, $CD_{p}$ is the compensation distance at the end with SAM in prediction, $CD_{e}$ is the compensation distance at the end with SAM in the experiment, $CD_{pL}$ = $CD_{p}/L$, and $CD_{eL}$ = $CD_{e}/L$.

## Data Availability

Not applicable.

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
