# Peer review of "Research on Self-Stiffness Adjustment of Growth-Controllable Continuum Robot (GCCR) Based on Elastic Force Transmission"

_biomimetics, 2023, doi:10.3390/biomimetics8050433_

Round 1

Reviewer 1 Report

The manuscript is well written and interesting. The methodology and descriptions are detailed, with good visualizations. I have a few comments regarding some parts of the manuscript:

1. Section numbering starts at 0 for introduction, this should be adjusted.
2. Figures 5 and 6 may benefit from adding dimensions to the schematic, or providing an approximate possible ranges of dimensions in figure descriptions, for easier understanding of the robot shape.
3. I believe that presenting the results visible in Figures 9 and 10 in a numerical manner, e.g. a table, with the average and maximum error (distance?) between the predicted and measured shape would be beneficial.
4. Related to the above results, it seems that there is correlation between the L and the error - as the longer shape seems to have a larger error. This should be commented on by the authors in more detail, and authors should see if there is a correlation that can be established.
5. Authors should more directly compare their results to Table 1, in conclusions or discussion.

Due to some additional work which is greater than simple minor corrections, I suggest major revisions to the article.

Author Response

Dear Professor,

We appreciate the reviewers and editors for many positive and constructive comments and suggestions. The main revisions made in this paper and responses to the reviewers’ comments are presented below. The comment response letter will contain the following illustrations and tables. 

Please see the attachment. Sincerely appreciate all your valuable comments and feedback.

Mingyuan Wang and Sheng Bao

Reviewer 2 Report

The paper proposes a stiffness-adjustable design for growing continuum robots. The proposed design is a rod-driven robot which is fed into the environment through linear stages, thus achieving follow-the-lead motion through model rather than passively. Stiffness modulation is achieved through a spring-loaded mechanism that increases reaction/elastic forces as the robot grows into the environment. The system is shown to have a good increase in stiffness when compared to a design without the proposed mechanism.

Overall, I found the paper interesting. I'd suggest the authors to address the following points before publications.

1) I find the design solution innovative but not well explained, as I had to reread through the paper several times before understanding it. I'd suggest the authors report Fig. 5 before the current Fig. 4, as I find the 2D clearer to start with, and, if possible, I'd appreciate more figures of the details of cable routing, possibly even with the actual prototype, for a better understanding of the system.

2) I'd suggest the authors to improve their introduction with specific reviews on stiffening in continuum and soft robotics [A-B], which are currently missing, as well as with more up-to-date reviews on continuum robots [C], which also addresses recent developments on stiffening.

[A] Manti, M., Cacucciolo, V., & Cianchetti, M. (2016). Stiffening in soft robotics: A review of the state of the art. IEEE Robotics & Automation Magazine, 23(3), 93-106.

[B] Yang, Y., Li, Y., & Chen, Y. (2018). Principles and methods for stiffness modulation in soft robot design and development. Bio-Design and Manufacturing, 1(1), 14-25.

[C] Russo, M., Sadati, S. M. H., Dong, X., Mohammad, A., Walker, I. D., Bergeles, C., Xu, K., & Axinte, D. A. (2023). Continuum robots: An overview. Advanced Intelligent Systems, 5(5), 2200367.

3) The authors should better discuss the advantage of their design when compared also to similar robots that can instrisically follow-the-lead thanks to extensible backbones. Two publications in particular [D, E] are in my opinion very similar to the proposed design. One of them [D] is already referenced as [26], and the second [E] is currently not in the bibliography.

[D] Kang, B., Kojcev, R., & Sinibaldi, E. (2016). The first interlaced continuum robot, devised to intrinsically follow the leader. PloS one, 11(2), e0150278.

[E] Amanov, E., Nguyen, T. D., & Burgner-Kahrs, J. (2021). Tendon-driven continuum robots with extensible sections—A model-based evaluation of path-following motions. The International Journal of Robotics Research, 40(1), 7-23. 

4) The authors propose a Piecewise Constant Curvature (PCCK) model to describe their system. As well known, this introduces significant errors, especially for longer continuum robots. Can the authors better estimate the relative positioning error (or its average deviation) from the actual position during the experiments? It would be interesting to revise Fig. 9 with the results from a more accurate model (e.g. ROM, Cosserat, or similar [C]) to see how much of the error is only due to PCCK.

5) The authors report results in loading only as fig. 11. Would it be possible to obtain more information on the robot behaviour at unloading? Usually, a large hysteresis is observed (see, for example, the experiments in [F]), and it would be interesting to know the effect of the proposed stiffening method on hysteresys and creep as well, as stiffening is shown to affect these behaviours significantly [F].

[F] Russo, M., Sriratanasak, N., Ba, W., Dong, X., Mohammad, A., & Axinte, D. (2021). Cooperative continuum robots: Enhancing individual continuum arms by reconfiguring into a parallel manipulator. IEEE Robotics and Automation Letters, 7(2), 1558-1565.

Author Response

(The authors gave the same response as above.)

Reviewer 3 Report

The manuscript reports a stiffness adjustment mechanism for growth-controllable continuum robot to improve the motion accuracy in variable scale motion. The stiffness adjustment is materialized by antagonism through cable force transmission during the length change of the continuum robot. Moreover, a statics model considering gravity and end load is established and evaluated. The experimental validations showed that the maximum error ratio is within 8.6%. Based on the reported results, the stiffness of the robot can be improved by 80%. The topic is very interesting and perfectly fits the scope of the journal. The manuscript is well-written and well-organized in general, and the quality of the graphics and language are also good. The following points should be addressed before consideration for the publication:

1.     Please check the numbering of the sections, for example: 0. Introduction!

2.     It is recommended that all the keywords should be mentioned in the abstract, for example "rod-driven continuum robot" is not mentioned in the abstract. Moreover, "statics model" and "mechanical design" are too general, please update accordingly.

3.     While a good and structured literature review is provided, more insights, both for conceptualization and application can be gained from other fields. The following references can be consulted as examples:

https://doi.org/10.1016/j.compstruct.2021.114976

https://doi.org/10.1088/1361-665X/aa8cb8

4.     The first-person point of view (for example in P3L91, P3L99, etc.) is not encouraged in academic writing. It is recommended to update the entire manuscript accordingly.

5.     In this reviewer idea, the discussed concept is novel, however the novelty is not highlighted in the manuscript, please update and enrich the manuscript (for example in Section 1 or 2) to reflect the research gap and the main contribution of the current study more clearly.

6.     Segmentation (compartmentalization) concept, as ultimate stiffness adjustment strategy, can be discussed somewhere in the manuscript, either as literature review or future perspective. In the following reference, the bio-inspired concept and its structural/mechanical application are discussed:

https://doi.org/10.3390/biomimetics8010095

7.     Section 6 (Discussion) just summarized the manuscript without any in depth discussion. It is more or less similar to the next section. In this situation, it is recommended to merge it with Section 7 (Conclusion). Alternatively, the author can keep this section by updating and enriching.

8.     Future needs briefly touched on in Sections 6 and 7. However, the discussion on future needs, open questions and possible improvements can be provided in a more detailed manner.

9.     Please very carefully check the references list, you will find some issues to fix. For example, issue number is missed in several cases. In [35]; missing journal name and details. In [22]: "In Proceedings of the Proceedings of", and etc.

Author Response

(The authors gave the same response as above.)

Round 2

Reviewer 1 Report

Respected Authors,

All of the comments I have provided have been fully addressed. I believe the manuscript may be accepted for publication at this point.

Kind regards.

Reviewer 3 Report

The manuscript is sufficiently improved and the authors have addressed satisfactory the reviewer's comments.